# Development of Fermented Rice Water to Improve the Quality of *Garaetteok*, a Traditional Korean Rice Cake

**DOI:** 10.3390/foods12030642

**Published:** 2023-02-02

**Authors:** Eun-Hyeong Lee, Hyun-Mo Jeong, Eun-A Kim, Ye-Rim Lee, Jae-Hoon Shim

**Affiliations:** Department of Food Science and Nutrition and The Korean Institute of Nutrition, Hallym University, Hallymdaehak-gil 1, Chuncheon 24252, Gwangwon-do, Republic of Korea

**Keywords:** fermentation, washed rice water, lactic acid bacteria, rice cake, retrogradation

## Abstract

In the rice processing industry, wastewater is an inevitable by-product of rice washing. To increase the utilization of washed rice water (WRW), seven types of fermented washed rice water (FWRW) were prepared using lactic acid bacteria (LAB) and carbohydrate hydrolase. The total concentration of small maltooligosaccharides (MOSs) in the amyloglucosidase (AMG) treatment groups was about ten times higher than in the untreated groups. After 6 h of fermentation, six of the seven FWRW samples reached a pH of 4 due to the increased concentration of organic acids and could, therefore, be used as food acidity regulators. To confirm the applicability of FWRW, the traditional Korean rice cake *garaetteok* was prepared with FWRW and stored at 4 °C for 5 days. A texture profile analysis (TPA) revealed that the hardness of *garaetteok* treated with FWRW was significantly lower than that of untreated *garaetteok* following storage. Differential scanning calorimetry (DSC) showed that FWRW retarded the retrogradation of *garaetteok* during storage. The addition of FWRW using *Lactobacillus reuteri* with an AMG group was particularly effective for inhibiting microbial activity in *garaetteok* during storage. These results suggest that FWRW using AMG-added *L. reuteri* can be used as a novel food additive for improving the quality of traditional Korean starch foods and could also reduce the volume of waste WRW.

## 1. Introduction

Rice is generally washed before cooking, with the resulting wastewater (washed rice water, or WRW) being discharged to the environment [1]. Washed rice water refers to the white, turbid water produced when washing rice [2,3,4,5,6]. Some WRW is consumed as food, but it is generally discarded during the food preparation process and is recognized as a by-product of rice production [7]. Washed rice water is a major pollutant in domestic sewage. About 85 million tons of WRW are produced per year in association with domestic rice consumption in Korea, and its management is estimated to cost about 15 billion US dollars [8,9]. In general, it takes about 440 times more water than the typical amount of discarded household sewage to make contaminated household sewage usable [9]. Washed rice water contains many nutrients that are beneficial to the human body, approximately half of which are carbohydrates (although it is also rich in protein and fat) [9,10]. Despite the presence of various physiologically active substances, there have been few attempts to develop foods using WRW, and it has generally been considered a by-product of rice production [2,3]. Recently, various studies have assessed the fermentation of rice water [5,6,10]. Using low-cost wastes, such as WRW, as a base reduces manufacturing costs and aids waste management [6,11]. Therefore, novel methods for the utilization of waste WRW are urgently required [12].

One option is to use WRW as a lactic acid bacteria (LAB) fermentation medium. Lactic acid bacteria have the potential to be used as probiotics in food [13]. They are non-spore Gram-positive bacteria that produce various organic acids (e.g., acetic and lactic acids) through the carbohydrate fermentation metabolism [14]. Some LAB have been used as probiotics and are designated as “generally recognized as safe” (GRAS) by the United States Food and Drug Administration [15]. Lactic acid bacteria have been extensively studied due to their economic importance to the food industry. They can produce various organic acids, including lactic and acetic acids, as well as natural food additives, such as flavor and texture enhancers [16,17]. They promote the digestion of lactose, benefit the intestinal immune system, and can prevent and treat diarrhea [18]. Lactic acid bacteria belong to one of the most important groups of microorganisms used in the food and biotechnology industries and are typically used as starter cultures in the production of dairy products and fermented plant-derived products [15,19]. Among the useful LAB metabolites, enzymes including carbohydrate-hydrolyzing enzymes, which are responsible for metabolic functions during the fermentation of LAB using WRW, delay the aging of carbohydrates in food [20]. In addition, organic acids (e.g., lactic and acetic acids) produced during lactic acid fermentation reduce pH and inhibit the growth of bacteria by affecting the electron transport system [21]. Among the organic acids produced by LAB, lactic acid and acetic acid are food additives that function as antioxidants, acidity regulators, and preservatives [22,23].

Rice cake is a traditional Korean food made by soaking grains, such as non-glutinous and glutinous rice grains, in water, followed by steaming, beating, roasting, or boiling. Rice cake is the main starchy food consumed in Asian countries [24,25,26]. Currently, rice cake accounts for about 50% of all processed rice products in Korea; white rice cakes are the most widely consumed variety [27,28]. During the COVID-19 pandemic, as in other countries, the frequency of dining out in Korea decreased among the general population, and the use of home meal replacement (HMR) services increased substantially. Compared to 2017, the sales of *garaetteok* increased by 24.5% in 2019 due to the decrease in dining out and the high spoilage resistance of processed foods, with *garaetteok* accounting for the largest proportion of processed rice products sold in Korea [29]. *Garaetteok* is a long cylindrical rice cake, which is prepared by gelatinizing water and salt in rice flour [30]. It is relatively easy to cook and, thus, easy to commercialize [31]. Nevertheless, the shelf life of bar rice cakes is <1 month due to the high potential for microbial contamination during the production process [27]. This problem not only makes it difficult to process the *garaetteok* stock but has also restricted its entry into the global market [32]. 

Similar to most processed starchy foods, in processed rice products, the moisture content is reduced during storage, and an irreversible crystal structure forms through the rearrangement of the side chains of starch, which is called retrogradation [33]. This starch-aging process decreases the quality of starchy foods, such as rice cakes, and affects their texture, taste, and merchantability [33]. Therefore, research on *garaetteok* has mainly focused on how to delay retrogradation [33,34]. Several recent studies have attempted to compensate for the low nutritional value of *garaetteok* by adding various natural substances, including ginseng, curry, garlic, black rice, and *Enteromorpha intestinalis* powder [26,29,33,35,36].

In this study, we attempted to improve the quality of rice cake products by using discarded WRW, which would also alleviate the associated environmental problems. Rice water fermentation was carried out using various types of LAB. Then, during storage, texture analyses, microbial detection, and determination of the retrograde rate of *garaetteok* with fermented washed rice water (FWRW) were conducted to determine the feasibility of using fermented rice water as a food additive for quality improvement.

## 2. Materials and Methods

### 2.1. Chemicals and Materials

Rice flour was purchased from a market (Agricultural Corporation Soyanggang, Chuncheon, Republic of Korea), and de Man, Rogosa, and Sharpe (MRS) medium was obtained from BD Biosciences (Franklin Lakes, NJ, USA). Lactic acid, acetic acid, malic acid, maleic acid, citric acid, sodium acetate (NaOAc), trifluoroacetic acid (TFA), glucose (G1), maltose (G2), and maltotriose (G3) were purchased from Sigma-Aldrich (St. Louis, MO, USA). Maltotetraose (G4) was acquired from Tokyo Chemical Industry Co., Ltd. (Tokyo, Japan). Maltopentaose (G5), maltohexaose (G6), and maltoheptaose (G7) were acquired from CarboExpert Inc. (Daejeon, Republic of Korea). Acetonitrile (HPLC grade) was purchased from J.T. Baker Chemical Co. (Phillipsburg, NJ, USA). Amyloglucosidase (AMG 300L Brew Q, Amylase^®^ AG 300 L) and α-amylase (Termamyl 120 L) were obtained from Novozymes (Bagsvaerd, Denmark). One unit of both AMG and Termamyl was defined as the amount that hydrolyzes 1 μmol of glycosidic bond per minute.

### 2.2. Lactic Acid Bacteria Strains

The MRS medium was completely mixed to maintain a pH of 6.5. After mixing, the medium was autoclaved for 15 min at 121 °C. Seven strains, *Lactobacillus plantarum* (KCTC 3015), *Lactobacillus curvatus* (KCTC 3767), *Lactobacillus casei* (KCTC 3110), *Lactobacillus acidophilus* (KCTC 3164), *Lactobacillus brevis* (KCTC 3498), *Lactobacillus reuteri* (KCTC 3594), and *Leuconostoc mesenteroides* (KCTC 3505), were obtained from the Korean Collection for Type Cultures (KCTC) and grown at 37 °C in the MRS medium for 21 h. The lactic acid bacteria were stored frozen at −50 °C in MRS medium containing glycerol as a freeze-protecting agent. The absorbance of the culture medium was measured, and the initial number of cells was adjusted to be the same as at 600 nm.

### 2.3. Preparation and Fermentation of WRW

To prepare the WRW, 200 mL of distilled water was added to 100 g of rice in a beaker. The solution was mixed with a vortex mixer for 10 min, and the absorbance was measured at 660 nm using a spectrophotometer (Optizen POP, Mecasys Co., Ltd., Daejeon, Republic of Korea) [37]. After the measurement, the absorbance value of the WRW was determined following dilution by a factor of 5.5. The supernatant was collected and sterilized at 121 °C for 15 min. The LAB (1%, *v*/*v*) were inoculated in the WRW and cultured in a shaking incubator (at 40 °C). During the incubation, the pH of the fermented WRW was measured using a pH meter (LAQUAtwin-pH-22; HORIBA, Kyoto, Japan). In the sample to which the enzyme was added, 0.1 U/mL of the enzyme was added during the fermentation.

### 2.4. Determination of Organic Acid Content

The organic acid content of the WRW was determined by high-performance liquid chromatography (HPLC) after filtering a sample of the undiluted solution with a 0.2-μm syringe filter. For the HPLC analysis, the UltiMate™ 3000 RSLC nano system (Dionex, Sunnyvale, MA, USA) was used; the column was a Triart C18 column (5.0 μm, 4.6 mm × 150 mm, YMC, Kyoto, Japan); the mobile phase had a flow rate of 0.425 mL/min at 37 °C, and the solvent consisted of water (0.1% TFA) and 20 mM phosphoric acid. Quantitative and qualitative analyses of the lactic and acetic acids were conducted using standard products under the same conditions. In the analysis, 25 µL aliquots of the filtered sample were injected into the column and eluted with acidified water (0.1% TFA; A) and 20 mM phosphoric acid (B) at a flow rate of 1 mL/min. The multiple gradient conditions were as follows: 0–100% B for 0–10 min, 100% B for 10–20 min, and 100–0% B for 20–27 min. The detector (UltiMate™ 3000, Dionex) monitored the eluent at 210 nm.

### 2.5. Analysis of Small Maltooligosaccharides (MOSs)

The sample was diluted five times in distilled water, filtered with a 0.2-μm syringe filter, and analyzed with an ion chromatography (IC) system (ICS-5000 IC; Dionex). The ICS-5000 system was equipped with a Dionex AERS-4 mm suppressor, ICS-5000 chromatography/pulsed amperometric detector (ED40; Dionex), ICS-5000 gradient pump, AS auto-sampler, and CarboPac PA1 guard columns (4 mm × 50 mm and 4 mm × 250 mm; Dionex). For the analysis, 20 uL aliquots of the filtered sample were injected into the column and eluted with multiple gradients of 150 mM NaOH (A) and 150 mM NaOH containing 600 mM NaOAc (B) at a flow rate of 1 mL/min. A multiple gradient of B (150 mM NaOH containing 600 mM NaOAc) was used as follows: 10–30% for 0–10 min, 30–40% for 10–16 min, and 40–50% for 16–27 min.

### 2.6. Garaetteok Preparation

The rice cake was prepared as described by Kim [38]. We added 1% (*v*/*v*) of the FWRW (50 mL) as the solvent to 100 g of rice flour (Saerom Food Co., Incheon, Republic of Korea). The rice flour was then steamed for 20 min and mixed for 5 min to obtain the dough. After the mixing, a die with a diameter of 10 mm was mounted into a molding machine (Foshan Shunde Yimingjia Plastic Co., Ltd., Foshan, China). The sample was molded into a thin bar of rice cake using an extrusion molding machine (SutaKing, MC-N186, Buwon Electronics Co., Ltd., Daegu, Republic of Korea), cooled in cold water for 20 min, and packed.

### 2.7. Texture Analysis

The textural properties of the *garaetteok* were measured using a TA-XT plus texture analyzer (SMS Co., Ltd., Croydon, UK) after being stored at 4 °C for 1, 3, and 5 days. The *garaetteok* was cut into the shape of a cylinder with 1 cm in diameter and 1 cm in height, and the hardness was measured 10 times per sample. For the texture profile analysis (TPA), a two-cycle compression was performed with a tester equipped with a 35-N load cell. The *garaetteok* slices were analyzed with a cylinder probe with a diameter of 35 mm at a rate of 10 mm/s and a rate of 20% strain.

### 2.8. Differential Scanning Calorimetry (DSC)

The *garaetteok* was analyzed using the DSC 4000 system (Perkin Elmer, Waltham, MA, USA) after 1, 3, and 5 days of storage at 4 °C. Direct DSC measurements of the *garaetteok* samples were performed without pretreatment. The samples were obtained from the center of the *garaetteok* after the aging process. The *garaetteok* samples (10 mg) were weighed and completely sealed in aluminum pans. The pans were heated from 10 °C to 140 °C at a heating rate of 10 °C/min. An empty pan was used as a reference. The starch retrogradation was measured based on the enthalpy, which was calculated in the area under the endothermic peak between 40 °C and 75 °C.

### 2.9. Microbiological Analysis

A total of 5 g of the *garaetteok* sample was obtained with scissors and tweezers using an aseptic technique. The sample was sterilized at 121 °C for 15 min in an autoclave and cut into fine pieces. After transferring the 5 g of minced sample to a sterilized bag, 45 mL of physiological saline (0.85% NaCl) was added and allowed to mix into the pores for 3 min. The abundances of general bacteria, *Escherichia coli*, *Staphylococcus aureus*, fungi, and yeast were measured using Petrifilm (3M, St. Paul, MN, USA). The samples were continuously diluted with 0.1% sterile physiological saline and plated on 3M petri films, with the general bacteria and *E. coli* grown at 37 °C for 48 h, *S. aureus* grown for 24 h at 37 °C, and fungi and yeast grown at 25 °C for 120 h. All stages of the analysis were carried out under sterile conditions. After the culturing, the fungi were counted (30–300 colonies) and expressed in log CFU/g.

### 2.10. Physical Characteristics of Garaetteok 

The color, pH, and water content of the samples were determined after 1, 3, and 5 days of storage at 4 °C. The water content of the *garaetteok* was measured six times after treatment in a dry oven at 105 °C with heating and drying at atmospheric pressure. 

The surface color of the *garaetteok* was evaluated by a colorimeter (CR-400; Minolta, Osaka, Japan). After an adjustment, the chromaticity was determined on a standard white plate (lightness [*L**] = 94.44; redness [*a**] = −0.01; yellowness [*b**] = 2.65). Each sample was measured 10 times.

### 2.11. Statistical Analysis

All experimental results were expressed as the mean ± standard deviation of three repeated experiments. Additionally, *p*-values of < 0.05 were considered significant, and the data were analyzed using Duncan’s multiple range test, performed with the SPSS software (version 25.0; SPSS Inc., Chicago, IL, USA).

## 3. Results

### 3.1. The pH Changes of WRW during Fermentation

Table 1 shows the changes in the pH of the WRW during fermentation with various LAB at 40 °C for 6 h. For all of the LAB, there was a continuous decrease in the initial pH from about 6 to pH 4. Overall, there were no significant differences in the pH between the untreated and fermented liquids with the addition of the enzyme, except for the WRW fermented using *L. reuteri*. Among the seven species of LAB, five bacteria that could lower the pH to ≤4.5 were selected as strains for subsequent rice water fermentation experiments, and the change in pH was examined by fermenting with various AMG concentrations. As shown in Figure 1, the pH of the fermentation solution treated with the LAB and the AMG enzyme decreased more quickly than the fermentation with only one strain. In the case of the *L. reuteri* group, the pH of the fermentation solution decreased more quickly than in the other groups as the amount of AMG increased (Figure 1A).

### 3.2. Analysis of the Products of FWRW

Small amounts of MOSs produced during the LAB fermentation were measured using high-performance anion-exchange chromatography (HPAEC) (Figure 2). A relatively large amount of glucose was generated in all groups treated with the AMG enzyme. Unusually, the amounts of glucose and MOSs produced in the *L. reuteri* fermentation group were lower than in the other groups.

The amount of organic acids produced in the rice water during the LAB fermentation was measured by HPLC; it increased as the fermentation progressed in all of the groups. The proportion of lactic acid gradually increased in all of the groups, except for the *L. reuteri* group, in which heterolactic acid fermentation was performed (Figure 3). The organic acid production was the highest in the *L. plantarum* group until the 5-h reaction, at which point the organic acids were produced in the order of *L. acidophilus*, *L. plantarum*, and *L. reuteri* when the AMG was added.

Among the groups using these three bacteria, although FWRW fermented with *L. acidophilus* produced the largest amount of organic acid, a unique scent made it undesirable for application to rice cakes. Therefore, *L. plantarum* and *L. reuteri* were selected as the final fermentation bacteria, and FWRWs were produced using these bacteria. The WRW fermented using *L. plantarum* and *L. reuteri* were abbreviated to FWRW-P and FWRW-R, respectively; when the AMG enzyme was added to each fermentation solution, they were named FWRW-PA and FWRW-RA, respectively. 

### 3.3. Properties of FWRW as an Additive for Inhibiting Microbial Growth in Garaetteok

In general, the organic acids (and some metabolites) produced by LAB inhibit the growth of harmful microorganisms and improve the storage of foods [24,39]. We prepared rice cakes by adding various types of FWRW and analyzed the growth patterns of microorganisms during storage (Table 2). During a 5-day storage at 25 °C, *E. coli*, yeast, and *S. aureus* were not detected in any of the groups. In the treated rice cakes, the number of colonies detected was relatively small compared to the control, and in the case of the FWRW-RA treatment group, bacteria were not detected until the 4th day of storage. 

### 3.4. Properties of FWRW as an Antistaling Agent in Garaetteok

*Garaetteok* hardens over time and, thus, loses its value as a product. Hardness is one of the important texture parameters indicating the staling of starchy foods [40]. To confirm the antistaling effect of FWRWs, *garaetteok* samples treated with various FWRWs were stored at 4 °C, and the change in hardness of the rice cakes according to the storage period was determined using a texture analyzer. In the *garaetteok* samples to which FWRW was added, the hardness was significantly (*p* < 0.05) lower during the storage period compared to the untreated samples (control). After production, the hardness of all samples was about 4400 g/cm^2^, with no significant differences between the samples (data not shown). However, when comparing the hardness of each sample on the fifth day of storage, the hardness of the control increased significantly to 12,113.6 g/cm^2^, while the hardness of the FWRW-RA was the lowest at 9690.3 g/cm^2^. (Figure 4).

To analyze the staling of the *garaetteok* more accurately, DSC was used to measure the area of the endothermic peak between 40 °C and 70 °C, which is known as the retrogradation peak of starch [41]. During the 5-day storage period, the peak area of the *garaetteok* without FWRW treatment was 35.50 mJ/mg, while the samples treated with FWRW had lower values (Figure 5). The retrogradation peak area was the smallest for the *garaetteok* treated with FWRW-RA at 33.34 mJ/mg (*p* < 0.05).

### 3.5. Preparation of Garaetteok Treated with FRWR and Analysis of Its Physical Properties

To determine the applicability of FWRW as an additive for rice cake manufacturing, *garaetteok* was prepared with the addition of various types of FWRW. One-tenth of the water used to make the rice cakes was replaced by FWRW. The basic characteristics (e.g., moisture content and chromaticity) of the FWRW-treated *garaetteok* were then analyzed. 

In *garaetteok*, the moisture content is important for overall quality and antistaling properties. The moisture content also affects the texture and storage stability [41]. As shown in Figure 6, after 5 days of storage, the moisture content of the *garaetteok* treated with FWRW-RA was 47.6%. The addition of FWRW-P had no effect on the moisture content, but for FWRW-PA, in which the AMG enzyme was added during the fermentation, there was a significant difference in the moisture content compared to the control (*p* < 0.05). 

The chromaticity of the five types of samples, i.e., the control and the FWRW-P, FWRW-PA, FWRW-R, and FWRW-RA samples, was measured and expressed as *L**, *a**, and *b** values (Table 3). In *garaetteok*, *L** and *b** tend to decrease during storage [30]. This was observed in all sample types; a decrease of 7% in the control was seen over the 5-day period, while in the FWRW-P, FWRW-PA, FWRW-R, and FWRW-RA groups, the decreases were 7.9%, 14.8%, 1.6%, and 8.2%, respectively. The *b** value for FWRW-RA slightly increased, while in all other groups, it decreased during storage. In the control, *L** decreased by 27.6%, while in the groups treated with FWRW, *b** ranged from 10.76 to 11.69 after 5 days, similar to the initial *b** value of the control.

## 4. Discussion

Starchy foods are a basic source of carbon in the human diet. Since many Asians eat rice as their staple food, it is important to develop ways to increase the shelf life of rice-processed products and find new ways to utilize discarded rice water. 

Disposal of leftover rice wastewater after washing rice is not only an environmental problem but also a serious operational problem for the rice processing industry [4]. Rice water also contains large amounts of valuable active ingredients. Rice water can be used as a natural detergent, plant nutrient, and food flavor enhancer [2,3,42]. In our study, a natural acidity regulator that inhibited the growth of decomposing microorganisms was developed by fermenting discarded rice water using seven species of LAB. Sufficient carbon is required for microbial fermentation [43], and in the case of WRW, the AMG enzyme can be introduced to increase carbon utilization because most carbon sources are macromolecules, such as amylose. It was found that *L. reuteri* grew faster and produced more organic acids when AMG enzymes were added during fermentation, which is thought to compensate for the lack of extracellular carbohydrate hydrolytic enzymes in *L. reuteri* (Figure 1A). 

The pH of the rice water was reduced to ≤4.5 by the organic acids produced during the fermentation of Lactobacillus. Depending on the strain used in the fermentation, each FWRW had a unique aroma and taste. To minimize this effect, various *garaetteok* were prepared by adding FWRW in several different proportions, with 1% (*v*/*v*) of the water finally being replaced by FWRW (data not shown). As shown in Table 2, during the storage period, the number of general bacteria detected was relatively small in the *garaetteok* with rice water fermented with *L. reuteri*. As shown in Table 1, the lowest pH was obtained for the *L. reuteri* fermented with AMG, and it is believed that the low pH played a major role in reducing the number of bacteria in the rice cake. Generally, LAB produces low-grade fatty acids, antibiotics, and hydrogen peroxide, as well as organic acids, thus inhibiting the growth of pathogens and decaying bacteria [44]. In addition, *L. reuteri* is able to suppress the growth of various types of bacteria by producing antibacterial substances that are effective in human or animal intestines and are not affected by pH or substances such as pepsin and trypsin [45].

As shown in Figure 2, the concentration of glucose in the solution was significantly higher in the groups treated with AMG compared to the untreated groups, which indicates that AMG can hydrolyze the carbohydrates contained in WRW and provide glucose to LAB. An effect of excess glucose in FWRW was also apparent; the moisturizing properties reduced the hardness of *garaetteok* (Figure 4 and Figure 6). Compounds with a large number of hydroxyl groups, such as glucose, reduce water activity (Aw) through hydrogen bonds with free water in foods [46]. As shown in Figure 6, *garaetteok* cakes prepared by adding FWRW-PA and FWRW-RA, which were fermented by AMG treatments, had significantly higher water content than bar rice cakes manufactured by adding FWRW-P and FWRW-RA; this was also seen during the storage period. The moisturizing effects delayed hardening, which is one of the most important issues for consumers (Figure 4). 

Measuring the endothermal peak area using DSC is useful for analyzing starchy foods [47]. Figure 5 shows that the lowest retrogradation rate was obtained for the *garaetteok* treated with FWRW-RA, which is consistent with the results of the hardness experiment. The FWRW-P, FWRW-PA, and FWRA-RA treatments had almost the same retrogradation-delaying effects on the *garaetteok*. We speculated that the FWRWs delayed the retrogradation of *garaetteok* via antistaling enzymes produced by LAB and the sugars contained in the fermentation solution. Recently, an antistaling method using amylase derived from LAB was reported [48,49]. *Lactobacillus* sp. possess an enzyme that selectively hydrolyzes the amylose involved in the initial retrogradation of starch and delays aging by treating bread with a crude enzyme [50]. Interestingly, the DSC analysis revealed that the FWRA-R treatment had a smaller antistaling effect than the FWRA-RA treatment, despite the inclusion of AMG (which enhances glucose generation in WRW) (Figure 5). According to Woo et al. [48], the expression of antistaling amylase enzymes in *Lactobacillus* sp. can vary depending on the source and amount of carbon in the medium. Therefore, it was assumed that *L. reuteri* produces fewer antistaling enzymes in an environment with less glucose, resulting in the weaker retrogradation-inhibiting effect of FWRW-R. 

## 5. Conclusions

In this study, to increase the microbial safety and storability of bar rice cakes, rice water discarded during rice processing was fermented with various LAB and applied to *garaetteok*. The addition of FWRW-RA to the rice cakes resulted in a 15% slower retrogradation rate and 20% lower hardness than the control *garaetteok*, suggesting that the fermentation of rice water using *L. reuteri* can not only increase the shelf life of starchy foods but could also be an effective way to utilize the waste WRW generated in the rice processing industry.

## Figures and Tables

**Figure 1 foods-12-00642-f001:**
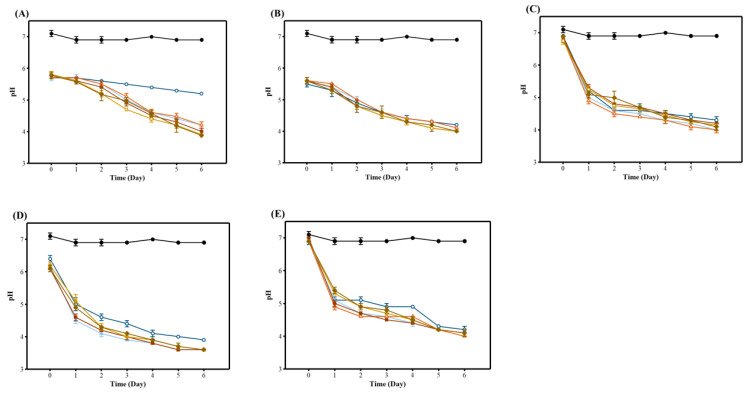
Changes in pH of FWRW with various AMG concentrations. Rice water was fermented using the following five types of LAB based on previous experiments: (**A**) *Lactobacillus reuteri*, (**B**) *Lactobacillus acidophilus*, (**C**) *Lactobacillus brevis*, (**D**) *Lactobacillus plantarum*, and (**E**) *Leuconostoc mesenteroides*. The pH was measured at 40 °C for 6 h. Closed circle: WRW; open circle: bacteria-treated WRW; closed triangle: 0.025 U/mL AMG-treated WRW; open triangle: 0.05 U/mL AMG-treated WRW; closed square: 0.1 U/mL AMG-treated WRW; open square: 0.15 U/mL AMG-treated WRW; closed diamond: 0.2 U/mL AMG-treated WRW.

**Figure 2 foods-12-00642-f002:**
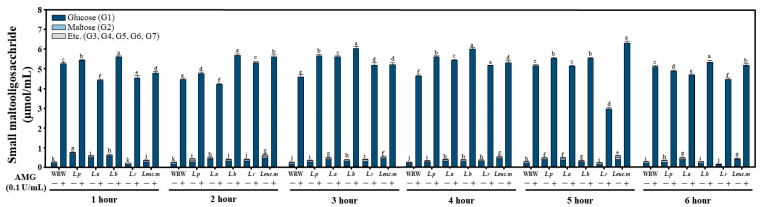
Analysis of the changes in the small MOS content during lactic acid fermentation. In all fermentation groups, the AMG enzyme-treated group had ≥10 times the total sugar content of the non-enzyme-treated group. Different letter in the same time zone means significantly different.

**Figure 3 foods-12-00642-f003:**
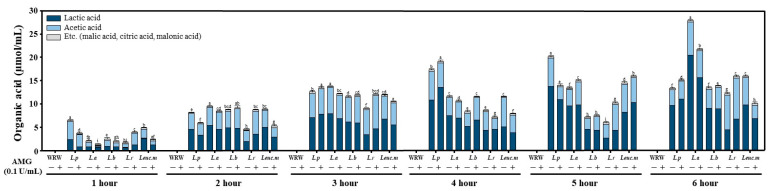
Analysis of changes in the organic acid composition during lactic acid fermentation. The organic acid concentration of the fermented rice water during lactic acid fermentation was analyzed every hour. In the enzyme-treated group, 0.1 U/mL of the treatment was applied. Different letter in the same time zone means significantly different.

**Figure 4 foods-12-00642-f004:**
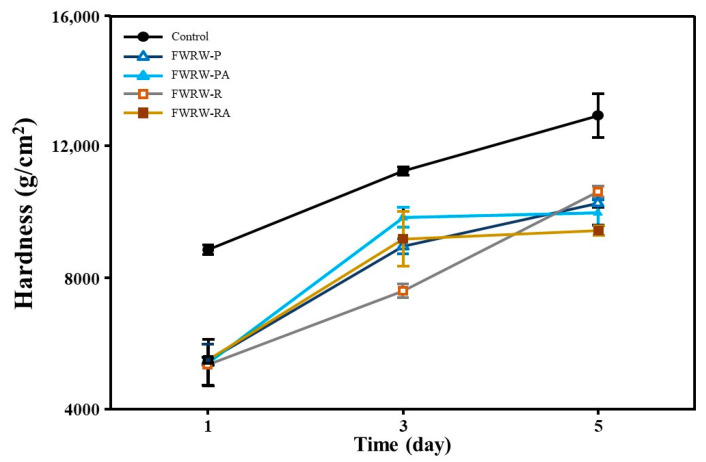
Changes in the hardness of FWRW-treated *garaetteok* during storage. Closed circle: control; open triangle: FWRW-P; closed triangle: FWRW-PA; open square: FWRW-R; closed square: FWRW-RA.

**Figure 5 foods-12-00642-f005:**
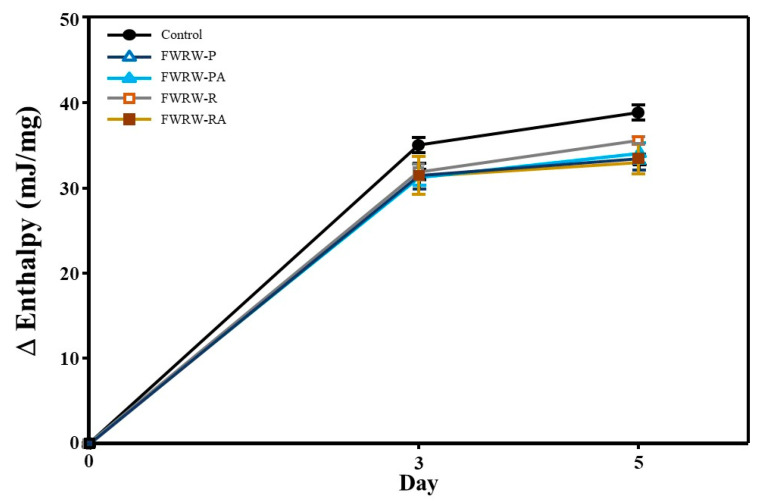
Retrogradation rates of the FWRW-treated *garaetteok*. Closed circle: control; open triangle: FWRW-P; closed triangle: FWRW-PA; open square: FWRW-R; closed square: FWRW-RA.

**Figure 6 foods-12-00642-f006:**
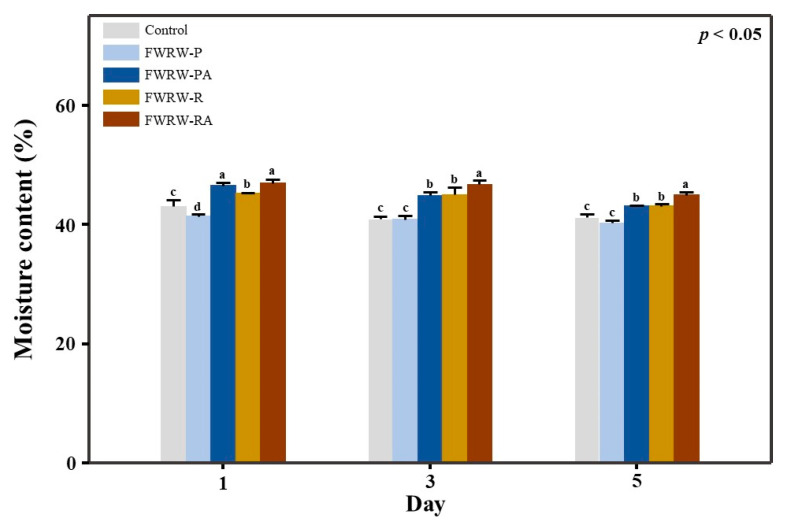
Changes in the water content of the FWRW-treated *garaetteok* during storage. The *garaetteok* was stored for 5 days at 4 °C, and the moisture content of the rice cake was measured. The FWRWs fermented using *L. reuteri* had more potent moisturizing effects than the FWRWs fermented using *L. plantarum*. Different letters in the same storage day mean significantly different.

**Table 1 foods-12-00642-t001:** Changes in the pH of fermented rice water according to the lactic acid bacteria and enzymes used in the treatments.

	Enzyme ^(1)^	Fermentation Time					
		1 h	2 h	3 h	4 h	5 h	6 h
*Lactobacillus acidophilus*	N	5.7 ± 0.1 ^aB^	5.4 ± 0.0 ^bA^	4.9 ± 0.1 ^cA^	4.5 ± 0.1 ^dB^	4.3 ± 0.1 ^eA^	4.3 ± 0.0 ^eA^
	AMG	5.8 ± 0.0 ^aB^	5.6 ± 0.1 ^bA^	5.0 ± 0.2 ^cA^	4.5 ± 0.1 ^dB^	4.3 ± 0.1 ^dA^	4.1 ± 0.1 ^eB^
	Termamyl ^(2)^	5.9 ± 0.1 ^aA^	5.6 ± 0.1 ^bA^	5.2 ± 0.1 ^cA^	4.6 ± 0.0 ^dA^	4.4 ± 0.0 ^eA^	4.3 ± 0.0 ^fA^
*Lactobacillus brevis*	NA	5.4 ± 0.0 ^aA^	5.4 ± 0.1 ^bA^	5.3 ± 0.0 ^bA^	4.8 ± 0.0 ^cA^	4.6 ± 0.0 ^dA^	4.3 ± 0.0 ^eA^
	AMG	5.4 ± 0.0 ^aA^	5.4 ± 0.1 ^aA^	5.3 ± 0.0 ^bA^	4.8 ± 0.0 ^cA^	4.4 ± 0.0 ^dB^	4.2 ± 0.0 ^eB^
	Termamyl	5.4 ± 0.1 ^aA^	5.4 ± 0.0 ^aA^	5.3 ± 0.1 ^bA^	4.9 ± 0.1 ^cA^	4.6 ± 0.0 ^dA^	4.3 ± 0.0 ^eA^
*Lactobacillus casei*	NA	5.6 ± 0.1 ^aB^	5.5 ± 0.1 ^aA^	5.0 ± 0.1 ^bA^	4.9 ± 0.1 ^bcA^	4.7 ± 0.1 ^cdA^	4.6 ± 0.1 ^eA^
	AMG	5.9 ± 0.1 ^aA^	5.6 ± 0.1 ^bA^	5.1 ± 0.1 ^cA^	4.9 ± 0.1 ^dA^	4.6 ± 0.1 ^eA^	4.4 ± 0.0 ^fB^
	Termamyl	5.8 ± 0.0 ^aA^	5.6 ± 0.0 ^bA^	5.1 ± 0.0 ^cA^	4.9 ± 0.1 ^dA^	4.7 ± 0.1 ^eA^	4.6 ± 0.0 ^fA^
*Lactobacillus curvatus*	NA	6.0 ± 0.1 ^aA^	5.9 ± 0.0 ^abA^	5.8 ± 0.0 ^bA^	5.6 ± 0.1 ^cA^	5.6 ± 0.0 ^dA^	5.5 ± 0.0 ^eA^
	AMG	6.0 ± 0.0 ^aA^	6.0 ± 0.0 ^aA^	5.9 ± 0.1 ^aA^	5.6 ± 0.1 ^bA^	5.5 ± 0.1 ^bcA^	5.5 ± 0.0 ^cA^
	Termamyl	6.1 ± 0.1 ^aA^	6.0 ± 0.1 ^abA^	5.9 ± 0.1 ^bA^	5.5 ± 0.0 ^cA^	5.6 ± 0.1 ^cA^	5.5 ± 0.1 ^cA^
*Lactobacillus reuteri*	NA	5.8 ± 0.1 ^aB^	5.6 ± 0.1 ^bA^	5.4 ± 0.1 ^cA^	5.3 ± 0.1 ^dA^	5.2 ± 0.0 ^dA^	5.2 ± 0.0 ^dA^
	AMG	5.9 ± 0.0 ^aB^	5.6 ± 0.1 ^bA^	5.2 ± 0.0 ^cB^	4.9 ± 0.1 ^dB^	4.6 ± 0.1 ^eC^	4.1 ± 0.1 ^fC^
	Termamyl	6.0 ± 0.1 ^aA^	5.6 ± 0.1 ^bA^	5.3 ± 0.1 ^cAB^	5.0 ± 0.1 ^dB^	4.7 ± 0.1 ^eB^	4.6 ± 0.0 ^fB^
*Lactobacillus plantarum*	NA	5.8 ± 0.0 ^aA^	5.4 ± 0.1 ^bA^	5.0 ± 0.0 ^cA^	4.6 ± 0.0 ^dA^	4.3 ± 0.0 ^eA^	4.1 ± 0.0 ^fA^
	AMG	5.5 ± 0.0 ^aB^	5.1 ± 0.0 ^bC^	4.6 ± 0.0 ^cC^	4.3 ± 0.0 ^dC^	4.3 ± 0.0 ^dA^	3.9 ± 0.0 ^eA^
	Termamyl	5.6 ± 0.0 ^aB^	5.2 ± 0.0 ^bB^	4.8 ± 0.0 ^cB^	4.5 ± 0.0 ^dB^	4.2 ±0.0 ^eA^	4.0 ± 0.0 ^fA^
*Leuconostoc mesenteroides*	NA	5.6 ± 0.1 ^aA^	5.5 ± 0.0 ^aA^	5.4 ± 0.1 ^aA^	5.0 ± 0.2 ^bA^	4.7 ± 0.1 ^cA^	4.4 ± 0.1 ^dA^
	AMG	5.6 ± 0.0 ^aA^	5.5 ± 0.0 ^aA^	5.4 ± 0.1 ^bA^	4.8 ± 0.0 ^cB^	4.4 ± 0.0 ^dB^	4.2 ± 0.1 ^eB^
	Termamyl	5.5 ± 0.0 ^aA^	5.5 ± 0.1 ^aA^	5.4 ± 0.0 ^bA^	4.8 ± 0.0 ^cAB^	4.6 ± 0.0 ^dA^	4.3 ± 0.0 ^eA^

NA, not added; AMG, amyloglucosidase. ^(1)^ All enzymes were added in amounts of 0.1 U/mL. ^(2)^ Termamyl = α-amylase. Different uppercase or lowercase letters mean that each of them is significantly different in the same column or row, respectively (*p* < 0.05).

**Table 2 foods-12-00642-t002:** Comparison of the microbes detected during storage under various FWRW treatments of *garaetteok*.

Microbe	Day	Control ^(1)^	FWRW-P	FWRW-PA	FWRW-R	FWRW-RA
Total plate count (log CFU/g)	1	5.87 ± 0.05 ^aA^	4.60 ± 0.12 ^aB^	3.36 ± 0.10 ^aC^	1.90 ± 0.07 ^aD^	ND ^bE^
	2	7.77 ± 0.04 ^bA^	7.62 ± 0.05 ^bB^	6.83 ± 0.08 ^bC^	6.61 ± 0.03 ^bD^	ND ^bE^
	3	8.92 ± 0.09 ^cA^	8.24 ± 0.08 ^cB^	7.65 ± 0.05 ^cC^	7.66 ± 0.04 ^cC^	ND ^bD^
	4	9.14 ± 0.03 ^dA^	8.81 ± 0.07 ^dB^	7.99 ± 0.09 ^dC^	8.75 ± 0.09 ^dB^	ND ^bD^
	5	9.41 ± 0.03 ^eA^	8.98 ± 0.02 ^eB^	9.00 ± 0.02 ^eB^	8.96 ± 0.03 ^eB^	5.41 ± 0.05 ^aC^
*E. coli* (log CFU/g)	1	ND	ND	ND	ND	ND
	2	ND	ND	ND	ND	ND
	3	ND	ND	ND	ND	ND
	4	ND	ND	ND	ND	ND
	5	ND	ND	ND	ND	ND
Yeast & Mold (log CFU/g)	1	ND	ND	ND	ND	ND
	2	ND	ND	ND	ND	ND
	3	ND	ND	ND	ND	ND
	4	ND	ND	ND	ND	ND
	5	ND	ND	ND	ND	ND
*S. aureus* (log CFU/g)	1	ND	ND	ND	ND	ND
	2	ND	ND	ND	ND	ND
	3	ND	ND	ND	ND	ND
	4	ND	ND	ND	ND	ND
	5	ND	ND	ND	ND	ND

ND, not detected. ^(1)^
*Garaetteok* treated with water only. Data are shown as means ± standard deviations (*n* = 3). Different lowercase or uppercase letters mean that each of them is significantly different in the same column or row, respectively (*p* < 0.05).

**Table 3 foods-12-00642-t003:** Comparison of the color changes of *garaetteok* under various FWRW treatments and storage periods.

	Storage Time (Days)	Control	FWRW-P	FWRW-PA	FWRW-R	FWRW-RA
*L** ^(1)^	0	69.06 ± 0.03 ^c^	71.25 ± 0.66 ^b^	72.40 ± 0.83 ^a^	67.73 ± 0.59 ^d^	68.14 ± 0.10 ^d^
	3	63.57 ± 0.05 ^d^	66.93 ± 0.05 ^bc^	62.78 ± 0.21 ^e^	66.64 ± 0.13 ^c^	70.62 ± 0.01 ^a^
	5	64.25 ± 0.90 ^b^	65.64 ± 0.03 ^a^	61.66 ± 0.12 ^c^	66.65 ± 0.58 ^a^	62.57 ± 0.64 ^c^
*a** ^(2)^	0	−1.28 ± 0.05 ^c^	−1.24 ± 0.01 ^c^	−1.76 ± 0.02 ^a^	−1.13 ± 0.02 ^d^	−1.56 ± 0.02 ^b^
	3	−2.15 ± 0.01 ^a^	−1.44 ± 0.00 ^c^	−1.12 ± 0.03 ^e^	−1.37 ± 0.04 ^d^	−1.92 ± 0.01 ^b^
	5	−2.02 ± 0.02 ^a^	−1.19 ± 0.00 ^e^	−1.51 ± 0.02 ^b^	−1.30 ± 0.01 ^d^	−1.34 ± 0.00 ^c^
*b** ^(3)^	0	11.71 ± 0.01 ^c^	12.82 ± 0.02 ^d^	11.07 ± 0.03 ^b^	11.04 ± 0.03 ^b^	10.42 ± 0.36 ^a^
	3	10.06 ± 0.04 ^a^	13.13 ± 0.06 ^d^	11.71 ± 0.06 ^b^	11.91 ± 0.06 ^c^	11.84 ± 0.02 ^c^
	5	8.48 ± 0.02 ^a^	11.26 ± 0.03 ^d^	10.54 ± 0.02 ^b^	10.76 ± 0.02 ^c^	11.69 ± 0.12 ^e^

^(1)^*L*ightness; ^(2)^*R*edness; ^(3)^*Y*ellowness. Data are shown as means ± standard deviations (*n* = 3). Different letters in the same row mean significantly different (*p* < 0.05).

## Data Availability

Data is contained within the article.

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
