# Peer review of "Development of Fermented Rice Water to Improve the Quality of Garaetteok, a Traditional Korean Rice Cake"

_foods, 2023, doi:10.3390/foods12030642_

Round 1

Reviewer 1 Report

A large amount of rice-washing water is produced and discharged to pullute the environment. In this manuscript, rice-washing water was fermented with food-borne lactic acid bacteria with amyloglucosidase, and then mixed with rice flour to make Garaetteok. The resulted products had improved texture. Its originality is high. The following items should be considered.

Line 134, 147: In 2.4 and 2.5, the type and condition of the corresponding detector should be given.

Line 162, 169: how to cut into a cube if a die with a diameter of 10 mm?

Line 173: a cylinder probe with a diameter of 35 mm?

Line 174: compression distances of two times?

Author Response

Dear Reviewer,

We are grateful for your insightful comments. Your recommendations were helpful and extended our knowledge regarding Texture profile analysis. 

The manuscript was modified according to your suggestion. Please see the attached file.

Best Wishes,

Q1. In 2.4 and 2.5, the type and condition of the corresponding detector should be given.

Ans. The information of corresponding detectors were properly added (line 127-128, line 133)

Q2. Line 162, 169: how to cut into a cube if a die with a diameter of 10 mm?

Ans. We found an error in the explanation. The cylindrical bar rice cake with a diameter of 1 cm was cut to a height of 1 cm (line 150-151)

Q3. Line 173: a cylinder probe with a diameter of 35 mm?

Ans. We found that our explanation was insufficient. The sentence was corrected properly (line 152-153).

Q4. Line 174: compression distances of two times?

Ans. The explanation regarding TPA was not clear. The method was added properly (line 152-153).

Reviewer 2 Report

The manuscript has investigated the possible application of fermented rice water to improve the quality of traditional Korean rice cake. The topic is interesting; however, the manuscript has several problems. 

1. I highly recommend mentioning the "garaetteok" in the title.

2. Section 2.1; Please provide the enzyme activity of the enzymes.

3. Section 2.3; Please explain more about the preparation and fermentation of WRW. The application of enzymes was not mentioned.

4. Please mention the model and the company name of the spectrophotometer and extrusion molding machine devices.

5. Please use L*, a*, and b*, instead of L, a, and b.

6. For better understanding, please use colored figures.

7. For Figures 2 and 3; Please add the standard deviation and the significant letters.

8. Lines 298-300; "The hardness was lowest for garaetteok treated with FWRW-RA at 9,690.3 g/cm2 on the 5th day of storage", From what?

Also, provide the hardness range for the control sample.

9. In the materials and methods section, the authors mentioned that they evaluated hardness, as well as cohesiveness, adhesiveness, chewiness, and springiness, but they did not present them in the results section!

Author Response

Dear Reviewer, 

We are grateful for your insightful comments. Your recommendations were helpful and extended our knowledge regarding rice cake analysis. 

The manuscript was modified according to your suggestion. Please see the attached file.

Best Wishes,

-----------------

Q1. I highly recommend mentioning the "garaetteok" in the title.

Ans. The title has been changed due to the reviewer's recommendation (line 2-3).

Q2. Section 2.1; please provide the enzyme activity of the enzymes.

Ans. The definition of enzyme activity was not sufficient. Therefore, we added the definition of unit (line 97-98).

Q3. Section 2.3; please explain more about the preparation and fermentation of WRW. The application of enzymes was not mentioned.

Ans. We agree that the information on enzyme addition was not clear. In the sample to which the enzyme was added, 0.1 U/mL of the enzyme was added during fermentation. The amount of enzyme was added to section 2.3 and associated tables and figure annotations (line 116)

Q4. Please mention the model and the company name of the spectrophotometer and extrusion molding machine devices.

Ans. The information of devices were added (line 111- 112 & line 145).

Q5. Please use L*, a*, and b*, instead of L, a, and b.

Ans. L, a, and b. were corrected properly (line 178-179, line 249, line 252-254, Table 3)

Q6. For better understanding, please use colored figures.

Ans. The figures were changed to color figures (Fig. 1 – Fig. 6).

Q7. For Figures 2 and 3; Please add the standard deviation and the significant letters.

Ans. The standard deviation and the significant letters were added (Fig. 2).

Q8. Lines 298-300; "The hardness was lowest for garaetteok treated with FWRW-RA at 9,690.3 g/cm2 on the 5th day of storage", From what? Also, provide the hardness range for the control sample.

Ans. We totally agree that the explanation was not sufficient. After production, the hardness of all bar rice cakes was about 4,400 g/cm2, so there was no significant difference in the hardness between samples, but as shown in Figure 4, it changed dramatically during the storage at 4°C. The manuscript was modified (line 229-232).

Q9. In the materials and methods section, the authors mentioned that they evaluated hardness, as well as cohesiveness, adhesiveness, chewiness, and springiness, but they did not present them in the results section!

Ans. We agree with the reviewer. Indeed, in the TPA system, hardness, cohesiveness, adhesiveness, and chewiness are analyzed together. We focused on the change in hardness of bar rice cake during storage. Therefore, Method 2.7 has changed according to the recommendation (line 151).

Round 2

Reviewer 2 Report

The manuscript is now acceptable.